# Enhancement of Cisplatin Cytotoxicity by Cu(II)–Mn(II) Schiff Base Tetradentate Complex in Human Oral Squamous Cell Carcinoma

**DOI:** 10.3390/molecules25204688

**Published:** 2020-10-14

**Authors:** Rasha H. Al-Serwi, Gamal Othman, Mohammed A. Attia, Eman T. Enan, Mohamed El-Sherbiny, Seham Mahmoud, Nehal Elsherbiny

**Affiliations:** 1Basic Dental Sciences, College of Dentistry, Princess Nourah bint Abdulrahman University, Riyadh 84428, Saudi Arabia; 2Oral Biology Department, Faculty of Dentistry, Mansoura University, Mansoura 35516, Egypt; 3Biochemistry Department, Faculty of Medicine, Mansoura University, Mansoura 35516, Egypt; jyounis@mcst.edu.sa; 4College of Medicine, Almaarefa University, Riyadh 71666, Saudi Arabia; matia@mcst.edu.sa (M.A.A.); msharbini@mcst.edu.sa (M.E.-S.); 5Department of Clinical Pharmacology, Faculty of Medicine, Mansoura University, Mansoura 35516, Egypt; 6Department of Pathology, Faculty of Medicine, Mansoura University, Mansoura 35516, Egypt; emanenan@mans.adu.eg; 7Anatomy Department, Faculty of Medicine, Mansoura University, Mansoura 35516, Egypt; 8Chemistry Department, Zagazig University, Zagazig 44519, Egypt; allahakbr16@yahoo.com; 9Department of Biochemistry, Faculty of Pharmacy, Mansoura University, Mansoura 35516, Egypt; 10Department of Pharmaceutical Chemistry, Faculty of Pharmacy, University of Tabuk, Tabuk 71491, Saudi Arabia

**Keywords:** schiff base, cisplatin, oral squamous cell carcinoma (SCCs), anticancer, apoptosis

## Abstract

Oral squamous cell carcinoma (SCC) is one of the most predominant tumors worldwide and the present treatment policies are not enough to provide a specific solution. We aimed to assess the cytotoxic effect of Cu(II)–Mn(II) Schiff base tetradentate complex alone or in combination with cisplatin against squamous cell carcinoma cell line (SCCs) in vitro. Oral-derived gingival mesenchymal stem cells (GMSCs) were used as control. The cell viability was assessed by MTT assay. IC50 values were calculated. Evaluation of apoptosis and DNA damage were performed. In addition, the expression of pro-apoptotic and anti-apoptotic genes and proteins were tested. IC50 values indicated less toxicity of the Schiff base complex on GMSCs compared to cisplatin. Schiff base complex treatment resulted in up-regulation of p53 and Bax genes expression and down-regulation of Bcl2 gene expression in SCCs paralleled with increased protein expression of caspase-3 and Bax and down-regulation of Bcl-2 protein. Annexin V-FITC apoptosis kit showed a higher apoptotic effect induced by a Schiff base complex compared to the cisplatin-treated group. These effects were markedly increased on the combination of Schiff base and cisplatin. The present study established that Cu(II)–Mn(II) Schiff base tetradentate complex might induce a cytotoxic effect on SCCs cells via induction of the apoptotic pathway. Moreover, this Schiff base complex augments the anticancer effect of cisplatin.

## 1. Introduction

Oral cancer is one of the public malignancies worldwide that accounts for 2–4% of all cancer cases, with nearly one million new cases diagnosed annually. Indeed, oral cancer ranks eighth among the most common causes of cancer-related deaths. Oral squamous cell carcinoma (SCC) is an aggressive global malignancy constituting more than 90% of diagnosed oral cancer cases [1]. Most cases occur in males over the age of 50, but the incidence in women and young patients is listed to be increasing [2]. Oral SCC is frequently detected in late advanced stages and has a poor prognosis with a 50–60% five-year survival rate [3]. Risk factors for oral SCC include age, gender, alcohol, tobacco, diet, and nutrition, with tobacco smoking being the most common risk factor [4].

Treatment strategies for oral SCC have significantly enhanced in the past few years; however, the survival rate is still poor, especially in advanced cases. Chemotherapy, radiotherapy, and surgery are currently conventional therapies for oral SCC. Treatment options vary mainly depending on the disease stage. Chemotherapy is usually advised for patient’s ineligible for surgery due to extensive local tumor growth, medical comorbidities, or distant metastasis. Chemotherapy has been used alone as a primary treatment modality or as a radiation sensitizer in combination with radiotherapy. Of note, combination therapy has been reported to increases the five-year total survival rate by 6.5% compared with a single modality treatment [5]. Indeed, combination therapy has been shown to be more effective in advanced oral SCC [6]. However, this efficacy was limited by high rate of recurrence, chemoresistance to some anticancer drugs, and harmful side effects [7].

Among current chemotherapies, cisplatin or cis-diamminedichloroplatinum (II) (CDDP) is a first-generation anticancer agent that has been commonly used for several years to treat oral SCC. Cisplatin triggers cytotoxicity through binding to DNA, the creation of DNA adducts, and interfering with the cell repair leading ultimately to apoptosis. However, the rapid development of both inherent and acquired chemoresistance to cisplatin restricts its clinical application [8]. So, recent studies are directed to find new therapeutic strategies, including combination drug regimens for oral SCC treatment [9].

The condensation reaction of primary amine and carbonyl compounds makes Schiff bases. Schiff bases have a wide range of biological activities comprising anti-inflammatory, antimicrobial, and anticancer activities [10]. Their DNA-binding and DNA cleavage activities made-up the use of these compounds as potential drugs in cancer treatment. The Schiff bases antitumor activity has been related to its ability to intercalate DNA and interfering with the role of the topoisomerases, that regulate the DNA topology during the cell division [11]. Schiff bases possess the ability to stabilize various metal ions in different oxidation states, forming complexes. Multiple studies have demonstrated mild to good cytotoxicity of Schiff bases against various malignant tumors. Metal inclusion in the complex has a significant impact on the anticancer effectiveness of the Schiff bases [12]. Specifically, Schiff base heterodinuclear Cu(II)–Mn(II) complex is a novel Schiff base metal derivative with reported therapeutic efficacy against liver and colon cancer [13].

Tumor-derived cell lines as model systems with resemblances to the original endogenous human tumors are suitable resources currently used as screening platforms for estimating the potential efficacy of anticancer therapeutics and understanding the underlying biological processes in carcinogenesis [14]. Consequently, cell lines derived from multiple tumor tissues are currently aiding in identification of the events linked to the development of cancer to help the generation and establishment of therapies alternative to the current defective modalities [15]. 

Stem cell research has become an advantageous and advanced scientific research field. Stem cell research pushes far beyond the therapeutic and scientific potential of regenerative medicine. Currently, the field of stem cell research is attracting enormous attention for application in drug screening [16]. Dental stem cells are a good source of adult stem cells that are merely available by tooth extraction or during the replacement of primary tooth. Oral-derived gingival mesenchymal stem cells (GMSCs) are self-renewing and easily accessible mesenchymal adult stem cells widely applied in oral and maxillofacial diseases [17].

The present study was designed to assess the anticancer activity of Cu(II)–Mn(II) Schiff base tetradentate complex (see Appendix A) alone and in combination with cisplatin against oral squamous cell carcinoma (SCC) cell line using GMSCs as control. This was performed through in vitro assessment of cytotoxicity, induced apoptosis and gene and protein expression of apoptotic proteins.

## 2. Results

### 2.1. Flow-Cytometry for Cell Characterization

Phenotypic cell-surface marker analysis of GMSCs showed that they were highly positive for the mesenchymal cell markers CD90 (92.9%) and CD105 (95.9%), and negative for the hematopoietic lineage marker CD34 (10.1%) marker, Figure 1. These results showed that the isolated cells were mostly non-hematopoietic MSCs.

### 2.2. Cytotoxicity Assay

The cytotoxicity and cell viability of the Schiff base complex and cisplatin on both GMSCs and SCC were assessed by using MTT assay. The recorded data revealed that an increase in percentage cell inhibition occurred along with increasing both cisplatin and Schiff base complex concentrations, Figure 2. Besides, the IC_50_ values obtained were 1 µg/mL for cisplatin and 250 µM for the Schiff base complex. These IC_50_ values were used for further studies to detect apoptosis and necrosis by flow cytometry and gene expression by quantitative real-time PCR.

### 2.3. Apoptosis Detection (Flow Cytometry Analysis)

In order to get more information about the mechanism of the cell-growth inhibitory effect of different concentrations of Cis and Schiff base complex (SB) on all studied groups, cell-cycle distribution by flow cytometry techniques was used. The ability of treatment to induce apoptosis was examined by using Annexin V-FITC apoptosis kit showed considerable increased early and late apoptosis in GMSCs treated cells in compared with untreated cells. According to our data, treated cells percentage in early and late apoptosis was 29.77% and 3.40% (SB), 28.33% and 2.63% (cis), 21.60% and 25.67% (SB and Cis) respectively. While these proportion for the untreated cells was 0.63% and 0.63% respectively. After treatment of the SSC with SB and/or Cis, the cells were investigated for the detection of early, late apoptosis and necrosis by flow cytometry. As illustrated in Figure 3 and Figure 4, Table 1, Table 2, Table 3 and Table 4, the rate of early apoptosis (Annx + /PI−) and late apoptosis (Annx + /PI+) in the cells treated with SB was calculated 27.83%, 4.50% and 3.40%, 55.20%, with Cis and 2.6%, 60.10% after SB and Cis respectively. However, untreated cells percentage in the early apoptosis and late apoptosis was 0.80 and 0.60. 

After treatment of GMSCs with SB and/or Cis, collected samples were analyzed for the determination of cell cycle phase by flow cytometry assay. For SB, Cis and combination, relative to G1/control, the identification of an increase of treated cells in sub-G1 phase (apoptotic cells) was most 21.80%, 46.93%, 65.90% for G2, G3 and G4 and 12.13% for control cells. Investigation of sub-G1 peak of cell cycle in treated GMSCs cell showed an increase in apoptosis. In general, our results showed that treated cells with Cis and both SB and Cis compared with the control group showed significant decrease at G0/G1. While, no change in G0/G1 phase population has been observed following SB treatment as compared to control. Our data showed significant decrease in S and G2/M phase percentage in treated groups. After treatment of SSC with SB and/or Cis, collected samples were analyzed for the determination of cell cycle phase by flow cytometry assay. Cell cycle analysis showed significant increase of G4 cells in sub-G1 phase related to significant decrease in G0/G1 population after SB and Cis treatment compared to control group. Moreover, for G2 and G3 compared to G1/control, inhibition of proliferation occurred at the G0/G1 stage of the cell cycle, where, a larger number of cells were found to be accumulated. The average histogram plot of cell-cycle analysis also indicated slightly change in the proportion of G2/M phase and a significant decrease in S phase percentage. According to our data, untreated cells percentage in the sub-G1, G0/G1, S and G2/M phases were calculated as 9.23%, 56.63%, 26.13% and 8.00 respectively. While these proportion for the cells treated were 17.20%, 70.50%, 5.33% and 6.97% for SB, 37.20%, 49.17%, 6.83% and 6.80% for CIS and 61.50%, 27.00%, 2.30% and 9.20% for SBandCis, respectively Figure 3 and Figure 4 and Table 1, Table 2, Table 3 and Table 4.

### 2.4. Reverse Transcriptase Polymerase Chain Reaction of BAX, Bcl2, and P53

P53, BAX, and Bcl-2 gene expression were normalized to the housekeeping GAPDH gene. A real-time PCR analysis of GMSCs and SCC cells treated with the estimated concentrations of the Schiff base complex and cisplatin showed a significant increase in p53 and BAX mRNA expression with more upregulation with Schiff base complex than cisplatin and the upregulation with cisplatin is increased more with the addition of Schiff base complex. On the other hand, the Schiff base complex and cisplatin combination showed a more significant decrease in Bcl-2 mRNA expression compared to cisplatin-treated than the Schiff base complex group, as shown in Figure 5.

### 2.5. Flow Cytometry for BAX, Bcl2 and Caspase-3

To evaluate apoptosis induced by SB and/or Cis in GMSCs and to determine the apoptotic route, treated cells were used to evaluate the levels of anti-apoptotic, pro-apoptotic and end stage apoptosis proteins using the flow cytometry technique. There was a significant up-regulated caspase-3 when cells treated with SB (35.46%), Cis (42.96), and SB and Cis (55.54%) compared to untreated GMSCs (21.48%). To investigate the possible involvement of the mitochondrial pathway in this process, the effect of the drugs was assessed on the level of Bax and Bcl-2 proteins. After SB and/or Cis treatment, the level of the pro-apoptotic Bax protein was significantly increased in SB, Cis and SB and Cis groups (31.48%, 44.86% and 60.42% respectively) compared to untreated GMSCs (24.70%). In addition, the level of the anti-apoptosis Bcl-2 protein was significantly decreased in groups treated with Cis and SB and Cis (21.52%, 17.60%) compared to untreated GMSCs (25.20%). Insignificant change in Bcl-2 protein was found in SB treated group compared to untreated GMSCs group (25.20%).

To determine the apoptotic route in SSC treated with SB and/or Cis, the levels of anti-apoptotic, pro-apoptotic and end stage apoptosis proteins using the flow cytometry technique were evaluated. There was a significant up-regulated caspase-3 when cells treated with SB (41.08%), Cis (53.40%) and SB and Cis (83.50%) compared to untreated SSC (14.60%). These results confirm the induction of apoptosis in SSC. To investigate the possible involvement of the mitochondrial pathway in this process, the effect of the drugs was assessed on the level of Bax and Bcl-2 proteins. After SB and/or Cis treatment, the level of the pro-apoptotic Bax protein was significantly increased in SB, Cis and SB and Cis groups (43.26%, 54.54% and 93.48% respectively) compared to untreated SSC. Moreover, the level of the anti-apoptosis Bcl-2 protein was significantly decreased in SSC groups treated with SB, Cis and SB and Cis (33.54%, 29.52% and12.92 respectively) compared to untreated SSC (54.60%). Figure 6 and Figure 7 and Table 5 and Table 6.

## 3. Materials and Methods

All experimental procedures were performed according to protocols of the Ethics Committee of the Faculty of Dentistry, Mansoura University, and the animal house unit of Nile Center for Experimental Research (NCER), Mansoura city, Egypt.

### 3.1. Schiff Base Cu(II)–Mn(II) Complex Preparation

Manufacture and characterization of heterodinuclear Cu(II)–Mn(II) Schiff base tetradentate complex was performed according to the process described by Dede et al. [18]. The prepared compound was then preserved in a light protected tube at room temperature until use. A stock solution of the Schiff base complex was prepared by dissolving 1 mg per 1 mL dimethyl sulfoxide (DMSO, Sigma-Aldrich, St. Louis, MO, USA). Then various concentrations were prepared by dilution in culture media RPMI-1640 (RPMI, Invitrogen, Gibco, Paisley, UK) and Dulbecco’s modified Eagle’s medium (DMEM, Invitrogen, Gibco, Paisley, UK) supplemented with 1% penicillin/streptomycin/amphotericin (PSA, Invitrogen, Gibco, Paisley, UK) and 10% fetal bovine serum (FBS, Invitrogen, Gibco, UK). 10% (*w*/*v*) stock solution of cisplatin (cis-Diammine-di-chloroplatinic (II), cis-DDP), (Medac, Wedel, Germany) was prepared in phosphate-buffered saline (PBS, Pan-Biotech, Aidenbach, Germany). Then, 1% in DMEM was used for dilution. For combination treatment, 250 µM Schiff base complex was combined with 1.0 µg/mL cisplatin.

### 3.2. Isolation and Culture of Gingival Margin-Derived Stem Cells (GMSCs)

All in vitro steps were performed at Nile Center for Experimental Research (NCER), Mansoura city, Egypt (Stem Cell Unit). Isolation of gingival stem cells was according to a protocol described by Zhang et al. (2009) [19]. Four pathogen-free albino rats weighing 80–150 g were euthanized and rat tissue samples were collected from clinically healthy gingiva. To separate the epithelial and lower spinous layer, the samples were treated aseptically and incubated with dispase (2 mg/mL; Sigma-Aldrich) overnight at 4 °C. Thereafter, the connective tissues were minced into 1–3 mm^2^ fragments and digested in sterile phosphate-buffered saline (PBS) containing 4 mg/mL collagenase IV (Worthington Biochemical) for 2 h at 37 °C. The samples were then grown in Dulbecco’s Modified Eagle’s medium Ham’s F-12 (DMEM-F 12) with 1% antibiotic (streptomycin, penicillin, and amphotericin) and 10% fetal bovine serum (FBS). The explants were maintained in the incubator (5% CO_2_ humidified air and 37 °C) for 14–21 days to reach 80% confluency. The cells were then trypsinized and subcultured (Figure 8a).

### 3.3. Oral Squamous Cell Carcinoma Cell Lines (SCC) and Culture

H357 (Human oral squamous cell carcinoma, tongue) cell lines were obtained from Sigma-Aldrich Co. (St. Louis, MO, USA). Eagle’s Dulbecco’s modified medium was used as a cell culture medium enhanced with 100 IU/mL penicillin, 10% fetal bovine serum, and 100 μg/mL streptomycin. The cells were cultured at 37 °C under a humidified atmosphere containing 5% CO_2_. Periodic examination of cultured cells was performed by examining the cultured cells at a 400× magnification under a light microscope without preliminary fixation using an Olympus microscope (New York Microscope Company, Hicksville, NY, USA), Figure 8b.

### 3.4. Cell Characterization of GMSCs

Flow cytometry was used for determination of the immunophenotype. Gingival cells (5 × 10^5^) were incubated at room temperature for 20 min in the dark with individual fluorescein isothiocyanate (FITC)-conjugated primary monoclonal antibodies in 100 μL PBS. CD45, CD90, CD105, and CD34 were used as primary antibodies (Abcam, Cambridge, UK). Cells were then diluted in 2 mL PBS/bovine serum albumin followed by centrifugation and resuspension in 200 μL of paraformaldehyde 4%. BD Accuri C6 flow cytometer and BD Accuri C6 software programs were used for acquisition and analysis (BD Biosciences, San Jose, CA, USA).

### 3.5. Study Groups

Cell culture samples were divided into the following four groups: Group I non-treated control, which was subdivided into Ia (control GMSCs) and Ib (control SCC). Group II Schiff base complex (SB) treated, which was subdivided into IIa (GMSCs with SB) and IIb (SCC with SB). Group III cisplatin (cis-DDP) treated, which was subdivided into IIIa (GMSCs with cis-DDP) and IIIb (SCC with cis-DDP). Group IV treated with both Schiff base complex and cisplatin, which was subdivided into IVa (GMSCs with both SB and cis-DDP) and IVb (SCC with both SB and cis-DDP). The cells were treated with different concentrations of cisplatin and Schiff base complex, and then the cytotoxicity was observed by MTT assay. IC50 values were calculated, and the effective dose was used for this study.

### 3.6. Cell Viability and Proliferation Analysis

Reduction of tetrazolium dye MTT [3-(4,5-dimethyl-thiazol-2-yl)-2,5-diphenyltetrazolium bromide] [20] by viable cells was used to assess proliferation rate in response to treatment (MTT cell growth assay kit, Funakoshi, Japan). Cells were seeded in 96-well plate (Falcon; Lincoln Park, NJ, USA) at a density of 2 × 10^3^ in culture media (DMEM containing 10% FBS). The plate was incubated for 48 h, then various concentrations of cisplatin (0.5, 1.0, 1.5, 2.0 and 2.5 µg/mL) and Schiff base complex (150, 300, 450, 600 and 750 µM) were added and kept for 48 h. Following treatment, 10 μL of MTT solution was added to each well, and the plate was left for a further 4 h [21]. The media was aspirated and acid isopropyl alcohol was added to each well to solubilize formazan crystals and the developed color was recorded at 550 nm was performed using microplate Reader (MTP300, Tokyo, Japan). Six cultured wells were used for each drug concentration, and the experiment was replicated 3 times. The 50% inhibitory concentration (IC50) was calculated using Graph Pad Prism Programming.

### 3.7. Annexin- V-FITC Binding Assay

Apoptotic cells were stained and counted by flow cytometry using Annexin V-FITC apoptosis detection kit (Sigma-Aldrich, St. Louis, MO, USA). Following treatment, cells from control and treated wells were washed with PBS two times and suspended at a concentration of 1 × 10^6^ cells/mL in a one-fold binding buffer. Thereafter, 500 μL of each cell suspension was mixed with 5 μL of Annexin V FITC Conjugate and 10 μL of propidium iodide solution. The tubes were then incubated for 10 min at room temperature, and the reaction was assessed by flow cytometry instrument (FACStar caliber, Becton Dickinson, San Jose, CA, USA) using a filter over 600 nm for PI detection and 488 nm excitation and 515 nm bandpass filter for FITC detection. Annexin-v-positive/PI-negative cells were considered as early apoptotic cells, whereas Annexin-v-positive/PI-positive cells were reflected as late apoptotic cells, and the percentage of each phase was calculated with the software Cell-Quest [22].

### 3.8. Apoptotic Makers (Caspase-3, Bcl-2 and Bax) Assay

Cellular detection of the apoptotic markers caspase-3, bcl-2 and Bax activity was determined by FCM employing the fluorescein isothiocyanate conjugated monoclonal antibodies (caspase 3 FITC, Rabbit anti- active caspase 3 cat. No, 559341), (PE Hamster Anti-Mouse Bcl-2 Set Cat No. 556537) and (Bax Monoclonal Antibody (6A7) Cat No. MA5-14003) according to the manufacturer instructions. Following treatment, cells from control and treated wells were washed with 2 mL PBS/BSA (Bovine serum albumin) then centrifuged at 2000 rpm for 5 min. The supernatant was discarded and the pellet was re-suspended in 100 µL of PBS. Ten µmicroliters of the 1st antibody were mixed well then incubated for 30 min at room temperature in dark. Cells were then washed with 2 mL PBS/BSA, centrifuged at 2000 rpm for 5 min and supernatant was discarded. Ten µmicroliters of conjugated IgG secondary antibody was added and then the tubes were incubated for 20 min. at room temperature in dark. Finally, cells were re-suspended in 200 µL of 4% par formaldehyde in PBS. The percentage of each apoptotic marker positive cells was measured by flow cytometry (FACScan; Becton Dickinson). Acquired data were analyzed by the use of Cell Quest software [23].

### 3.9. Detection of Bcl-2-Associated X Protein (Bax), B Cell Lymphoma 2 (Bcl2), and P53 Genes Expression by Reverse Transcriptase Real-Time–PCR

Control and treated cells were harvested, and total RNA was isolated using the RNeasy Mini Kit (Qiagen, Venlo, The Netherlands) following manufacturer’s protocol that encloses extra steps to remove genomic DNA. The quality and purity of isolated RNA were estimated spectrophotometrically at 260 nm, and the 260/280 nm ratios, respectively, using Nanodrop (Thermo Scientific, Waltham, MA, USA). 1 μg RNA from each sample was reverse-transcribed into cDNA using the Thermo Scientific cDNA Synthesis Kit (Promega, Leiden, The Netherlands).

RT-PCR primers for studied genes BAX, Bcl-2, and p53 were designed using Primer-BLAST software from the National Center for Biotechnology (Bethesda, MD, USA). Glyceraldehyde-3-phosphate dehydrogenase (GAPDH) was used as a reference gene. CFX96 RT-PCR system (Bio-Rad, Hercules, CA, USA) was used to run RT-qPCR experiments using the following reaction mixture: 1 mM of forwarding/reverse primer (Eurogentec, Seraing, Belgium), 200 mM of dNTPs, and 0.75 U of Taq Polymerase (Roche). The thermal cycle was set using the following parameters: initial denaturation step at 94 °C for 5 min, 35 cycles of denaturation (60 s each, at 95 °C), annealing at 55 °C for 60 s, and an elongation step at 72 °C for 45 s, followed by final extra elongation step at 72 °C for 10 min [24]. The sequences of studied primers and product size were depicted in Table 7.

### 3.10. Statistical Methods

Data was analyzed using Statistical Package for Social Science software computer program version 26 (SPSS, Inc., Chicago, IL, USA). Quantitative parametric data were presented in mean and standard deviation while quantitative non-parametric data were presented in median (IQR). One-way analysis of variance (ANOVA) followed by post-hoc Tukey was used for comparing quantitative parametric data while Kruskal–Wallis followed by post-hoc Dunn’s was used for comparing quantitative non-parametric data. A log dose response curve was established to calculate IC50 for CIS and Schiff base complex in SCC and GMSCs groups by using graph pad prism version 8.0. *p* value less than 0.05 was considered statistically significant.

## 4. Discussion

Oral SCC is one of the most common cancers that remains noncurable with the existing therapies, and the survival of patients still has been a significant problem. Indeed, advanced oral SCC is chemotherapeutic and radiation-resistant by the current regimes [25]. So, additional augmentations in the new chemotherapeutic agents for oral SCC might improve patient survival rates and outcomes. Indeed, the identification of novel therapeutic choices is an aim of interest. 

The response rate of oral SCC to cytotoxic agents has not reached acceptable levels. Moreover, both the tumor cells’ resistance and dose-related toxicity stay two of the utmost significant problems in the chemotherapy of oral SCC [26]. These drawbacks have pointed researchers to think about combination therapy, in which different chemotherapeutic agents are combined with other cytotoxic compounds that enable decreased chemoresistance and augmented anti-tumergenic effect. Among the available anticancer agents, cisplatin has been widely studied to be used in combination therapy. However, to design a well combinative chemotherapeutic regime, there must be a better focus on lower systemic toxicity along with enhanced cytotoxic effect.

Schiff bases have been discovered to show a wide variety of potential therapeutic purposes, including anticancer and antimicrobial effects. Schiff bases retain the high potential to inhibit carcinogenesis in various cancers and this reported effect can be enhanced with complexation. However, their antitumor activity mechanism is still in doubt [27,28,29]. Meanwhile, new evolving data support the hopeful anticancer effect of Schiff base compounds against numerous types of cancer in vitro, like renal cell carcinoma [12], prostate cancer [30], hepatocellular carcinoma [10], and colon cancer [31]. On the other hand, its effect on oral SCC hasn’t been widely studied. 

Therefore, the present study was designed to test the cytotoxic and chemosensitizing effects of heterodinuclear Cu(II)–Mn(II) Schiff base tetradentate complex on oral SCC to recognize its potential promising role in oral cancer therapy. We used heterodinuclear Cu(II)–Mn(II) Schiff base tetradentate complex alone and in combination with cisplatin against oral SCC cell line. We also used GMSCs as control.

MTT assay revealed the cytotoxic effect of the Schiff base and cisplatin on oral SCC. The IC50 value was calculated to be used for further experiments. Calculated IC50 values indicated less toxicity of Schiff base on both oral SCC and GMSCs. Moreover, Schiff base and cisplatin treatment led to the up-regulation of expression of p53 and Bax genes (pro-apoptotic), and down-regulation of Bcl2 (anti-apoptotic). These results explained data from flow cytometry, which indicated induction of apoptosis in treated cells compared to untreated controls.

Moreover, the combination of cisplatin with Schiff base demonstrated greater effects compared to sole treatment. Our data are in agreement with the data of Trávníček et al. [32] who reported the cytotoxic effect of Schiff base ligand against ovarian carcinoma. Moreover, DNA cleavage activity and apoptosis of cancerous cells were previously reported as a mechanism Schiff base cytotoxic effect. Schiff base treatment alone and in combination with cisplatin compared to cisplatin alone caused more significant DNA damage and higher % of apoptotic cells arrested in the G0/G1 phase, as showed by flowcytometric analysis. These data are consistent with earlier studies recording that these complexes arrested the cell cycle in the G0/G1 phase and encouraged tumor cell apoptosis through a reactive oxygen species (ROS)-mediated mitochondrial pathway. Indeed, Schiff base cytotoxicity toward the HeLa cell lines was 1.9–3.5-fold more potent than cisplatin [33].

It is usually accepted that initiation of apoptosis is the primary cytotoxic tool of most cancer chemotherapeutic agents, and deviations in the control of the apoptosis process can affect the sensitivity of malignant cells to multiple drugs. These chemotherapeutic drugs mediate their effect via modulating various cellular apoptotic proteins and regulators. Previously, various Schiff base metal complexes demonstrated cytotoxicity via triggering initiator and effector caspases. Both initiator and effector caspases play fundamental role in apoptotic process. Activated initiator caspases subsequently activate specific effector caspases leading to proteolytic degradation of intracellular proteins to mediate apoptosis [27]. Moreover, p53 is a well-known regulator of apoptosis, and its variable expression has been involved as a reason for insensitivity or resistance of tumor cells [34]. The p53 apoptotic pathway is frequently disturbed in most human cancer [35].

P53-mediated apoptosis pathway comprises modifications of numerous executive components in apoptotic machines. One of them is Bax protein that functions as an apoptotic activator. The expression of the Bax gene is regulated by the tumor suppressor P53 but, this pro-apoptotic protein is inhibited by Bcl-2 that is localized to the outer membrane of mitochondria, as it plays a central role in supporting cellular survival and preventing the actions of pro-apoptotic proteins [36]. Therefore, BAX/Bcl-1 balance is crucial for the process of cell death. Bulatov, 2018 et al. [37] suggested that a Schiff base complex anti-proliferative effect can be both p53-dependent and independent. However, the presence of p53 exacerbates the cytotoxic effect of the complex. Herein, gene and protein expression analysis results showed that the used Schiff base complex could efficiently promote cell apoptosis via up-regulating the expression of P53 and Bax gene, caspase-3 and Bax proteins and down-regulating Bcl-2 gene and protein. These results are supported by a recent study by Zhang et al. [38] who reported the cytotoxic effect of a Schiff base derivative on liver cancer cells via modulatory effect on apoptosis proteins.

Moreover, Song et al. [39] studied the potential cytotoxic effects of three Schiff base complexes and reported specific cytotoxicity of these compounds to A549 cancer cells. The reported cytotoxic effect was mediated via the induction of DNA damage and inhibiting DNA synthesis. Similarly, our data demonstrated that Schiff base complex induced DNA damage in oral SCC, and this effect was enhanced on Schiff base complex cisplatin combination.

## 5. Conclusions

Collectively, our data suggest that the Schiff base exhibited a cytotoxic effect on oral SCC cells through modulation of apoptosis and modifications of various apoptotic factors. Further, cisplatin and Schiff base complex cotreatment may add to sensitizing oral SCC cells to sole treatment. Combination of cisplatin with Schiff’s based produced less anti-proliferative effects and pro-apoptotic effects on GMSCs compared to SCC. However, additional studies involving generating the oral SCC xenograft mouse model must be conducted to support our results and to evaluate if the cytotoxic and chemosensitizing effect of Schiff base is outweighing its toxicity on normal cells.

## Figures and Tables

**Figure 1 molecules-25-04688-f001:**
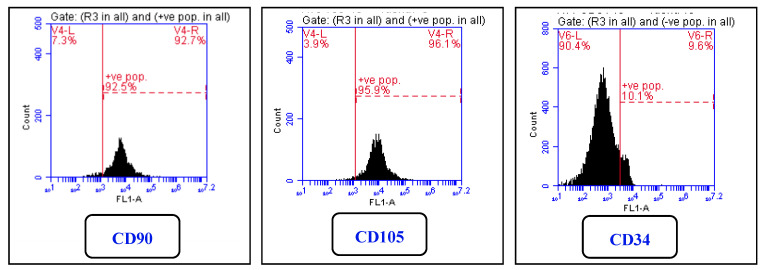
Histogram showing phenotypic cell-surface marker analysis of digested passage 3 (P3) Dental pulp stem cells (GMSCs). The GMSCs were highly positive for the mesenchymal markers CD90 and CD105 and were negative for the hematopoietic marker CD34.

**Figure 2 molecules-25-04688-f002:**
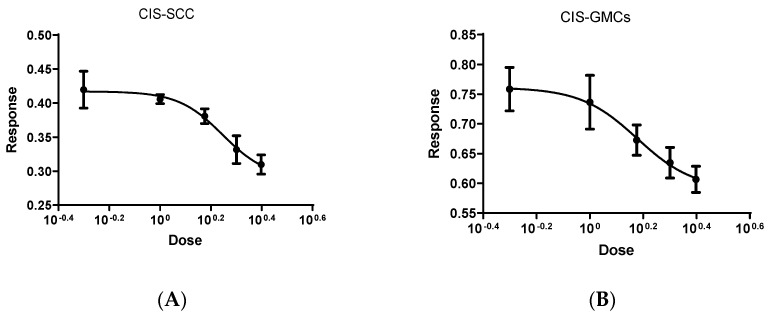
Evaluation of IC50 for different concentrations of Cis on SSC (**A**) and on GMCS (**B**). Evaluation of IC50 for different concentrations of Schiff-base on SSC (**C**) and on GMCS (**D**).

**Figure 3 molecules-25-04688-f003:**
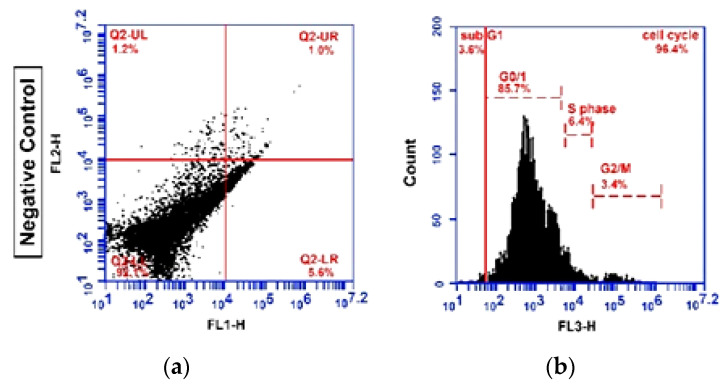
Flow cytometry analysis of negative control cells acquired without stain for: (**a**) annexin and (**b**) promidium iodide.

**Figure 4 molecules-25-04688-f004:**
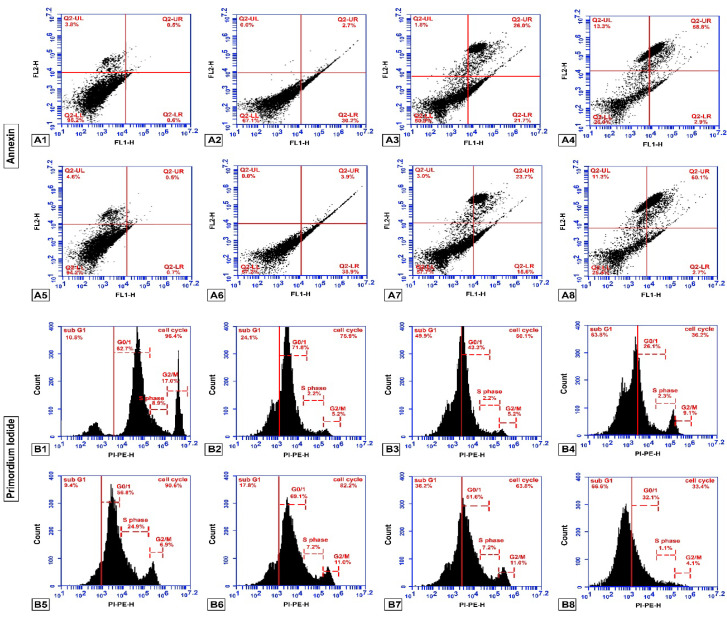
(**A**) Flow cytometry analysis of cell counts, detecting the number of annexin positive cells. (**B**) Flow cytometry analysis of cell counts, detecting the number of Propidium iodide (PI)-positive cells. 1: GMSCs control; 2: GMSCs treated with Schiff base complex; 3: GMSCs treated with cisplatin; 4: GMSCs treated with Schiff base complex and cisplatin together; 5: SCC cells; 6: SCC treated with Schiff base complex; 7: SCC treated with cisplatin; 8: SCC treated with Schiff base complex and cisplatin together.

**Figure 5 molecules-25-04688-f005:**
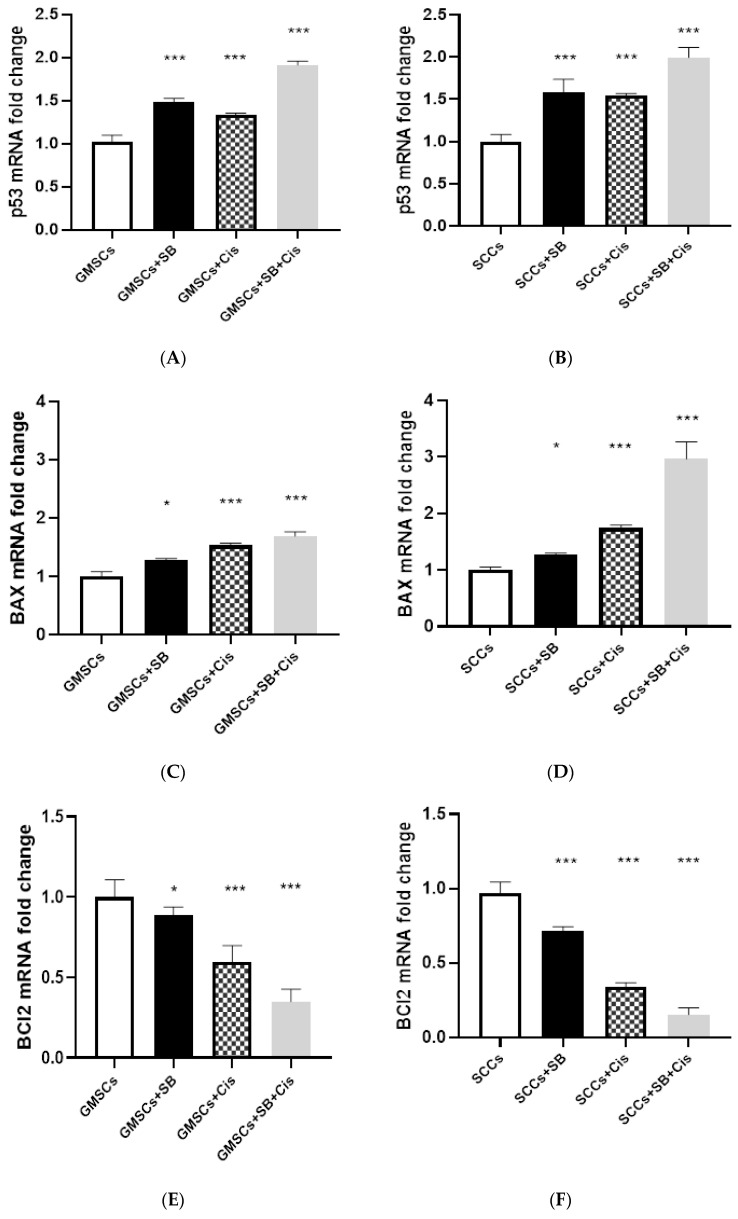
Effect of Schiff base complex (SB) treatment and cisplatin (Cis) on (**A**) p53 mRNA level on gingival mesenchymal stem cells (GMSCs); (**B**) p53 mRNA level on Oral squamous cell carcinoma (SCC); (**C**) BAX mRNA level on GMSCs; (**D**) BAX mRNA level on SCC; (**E**) BCl2 mRNA level on GMSCs; (**F**) BCl2 mRNA level on SCC. * *p* < 0.05; *** *p* < 0.001.

**Figure 6 molecules-25-04688-f006:**
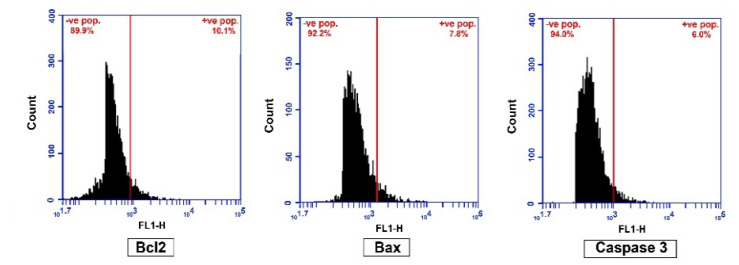
Flow cytometry analysis of negative control cells acquired without stain for Bcl2, Bax and Caspase 3.

**Figure 7 molecules-25-04688-f007:**
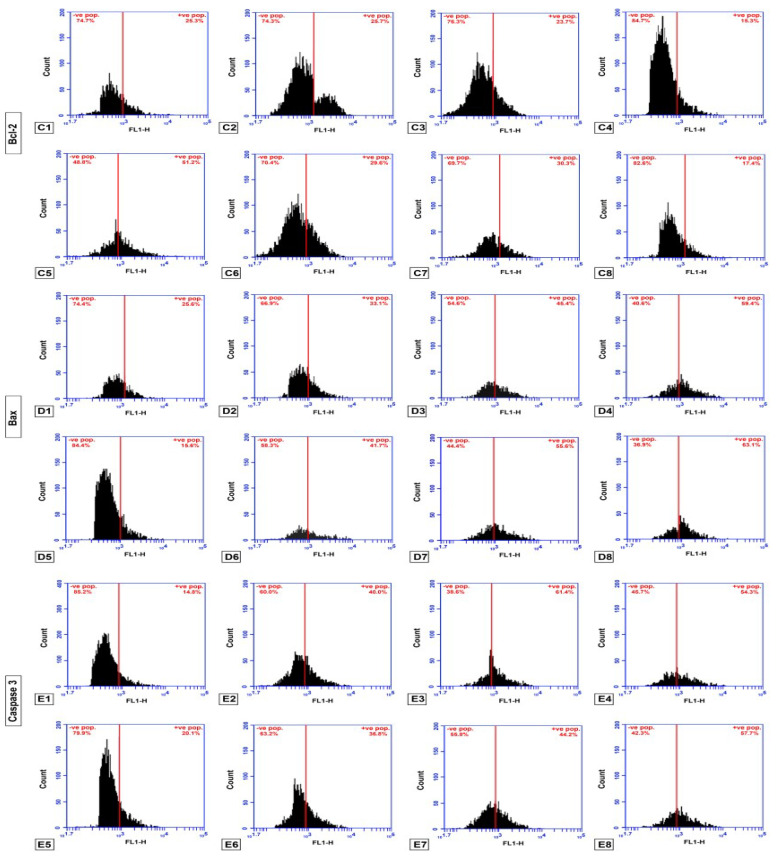
(**C**) Flow cytometry analysis of cell counts, detecting the number of Bcl-2 positive cells. (**D**) Flow cytometry analysis of cell counts, detecting the number of Bax positive cells. (**E**) Flow cytometry analysis of cell counts, detecting the number of Caspase 3 positive cells. 1: GMSCs control; 2: GMSCs treated with Schiff base complex; 3: GMSCs treated with cisplatin; 4: GMSCs treated with Schiff base complex and cisplatin together; 5: SCC cells; 6: SCC treated with Schiff base complex; 7: SCC treated with cisplatin; 8: SCC treated with Schiff base complex and cisplatin together.

**Figure 8 molecules-25-04688-f008:**
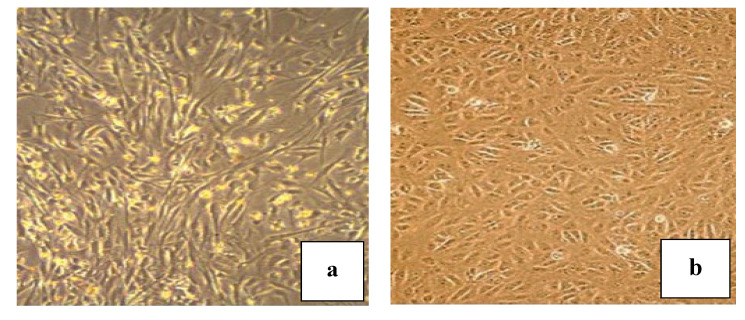
(**a**) cultured gingival mesenchymal stem cells (GMSCs), (**b**) cultured squamous cell carcinoma (SCC) cells, scale bars = 400 μm.

**Table 1 molecules-25-04688-t001:** Statistical analysis of annexin positive staining detected by flow cytometry in GMSCs.

	Control	SB	Cis	SB and Cis	*p* Value
Viable	95.43 ± 0.44	66.10 ± 0.85 ***	67.13 ± 0.76 ***	50.60 ± 0.80 ***^###¶¶¶^	<0.001
Early apoptosis	0.63 ± 0.05	29.77 ± 0.52 ***	28.33 ± 1.65 ***	21.60 ± 1.07 ***^###¶¶¶^	<0.001
Late apoptosis	0.63 ± 0.14	3.40 ± 0.86 ***	2.63 ± 0.45 ***	25.67 ± 0.52 ***^###¶¶¶^	<0.001
Necrotic	3.60 (2.50–3.80)	1.00 (0.00–1.20) ***	2.40 (0.00–3.30) *	1.80 (1.50–3.10)	0.003

Data expressed as mean ± SD and median (IQR). Test used: One-way ANOVA followed by post-hoc Tukey for data expressed as mean ± SD and Kruskal–Wallis followed by post-hoc Dunn for data expressed as median (IQR). * *p* < 0.05; *** *p* < 0.001 vs. control group. ^###^
*p* < 0.001 vs. SB Group. ^¶¶¶^
*p* < 0.001 vs. Cis group.

**Table 2 molecules-25-04688-t002:** Statistical analysis of annexin positive staining detected by flow cytometry in SCC.

	Control	SB	Cis	SB and Cis	*p* Value
Viable	93.77 ± 0.52	45.30 ± 14.59 ***	26.73 ± 0.88 ***^##^	25.50 ± 1.25 ***^##^	<0.001
Early apatosis	0.80 ± 0.09	27.83 ± 19.11 ***	3.40 ± 0.15 ^##^	2.63 ± 0.21 ^##^	<0.001
Late apaptosis	0.60 (0.50–2.10)	4.50 (3.9–56.80)	55.20 (55.20–57.10) **	60.10 (58.00–61.20) ***^##^	<0.001
Necrotic	4.60 (3.90–4.60)	2.10 (0.00–13.30)	14.00 (14.00–14.10) **^##^	12.40 (11.30–12.60)	0.001

Data expressed as mean ± SD and median (IQR). Test used: One-way ANOVA followed by post-hoc Tukey for data expressed as mean ± SD and Kruskal–Wallis followed by post-hoc Dunn for data expressed as median (IQR) ** *p* < 0.01; *** *p* < 0.001 vs. control group. ^##^
*p* < 0.01 vs. SB Group.

**Table 3 molecules-25-04688-t003:** Statistical analysis of PI positive staining detected by flow cytometry in GMSCs.

	Control	SB	Cis	SB and Cis	*p* Value
Sub G1	12.13 ± 0.80	21.80 ± 0.82 ***	46.93 ± 2.96 ***^###^	65.90 ± 0.56 ***^###¶¶¶^	<0.001
G 0/1	69.33 ± 2.01	68.27 ± 1.39	44.20 ± 0.89 ***^###^	30.30 ± 1.00 ***^###¶¶¶^	<0.001
S	9.03 ± 1.39	5.07 ± 1.61 **	4.10 ± 2.64 ***	2.00 ± 0.76 ***	<0.001
G2/M	9.50 ± 0.45	4.87 ± 1.90 ***	4.77 ± 0.52 ***	1.80 ± 0.85 ***^##¶¶^	<0.001

Data expressed as mean ± SD. Test used: One-way ANOVA followed by post-hoc tukey. ** *p* < 0.01; *** *p* < 0.001 vs. control group. ^##^
*p* < 0.01; ^###^
*p* < 0.001 vs. SB Group. ^¶¶^
*p* < 0.01; ^¶¶¶^
*p* < 0.001 vs. Cis group.

**Table 4 molecules-25-04688-t004:** Statistical analysis of PI positive staining detected by flow cytometry in SCC.

	Control	SB	Cis	SB and Cis	*p* Value
Sub G1	9.23 ± 0.23	17.20 ± 0.95 ***	37.20 ± 1.75 ***^###^	61.50 ± 1.12 ***^###¶¶¶^	<0.001
G 0/1	56.63 ± 0.27	70.50 ± 2.38 ***	49.17 ± 1.20 ***^###^	27.00 ± 0.91 ***^###¶¶¶^	<0.001
S	26.13 ± 1.60	5.33 ± 0.52 ***	6.83 ± 0.58 ***^#^	2.30 ± 0.18 ***^###¶¶¶^	<0.001
G2/M	8.00 ± 1.84	6.97 ± 2.60	6.80 ± 2.48	9.20 ± 0.09	0.17

Data expressed as mean ± SD. Test used: One-way ANOVA followed by post-hoc Tukey *** *p* < 0.001 vs. control group. ^#^
*p* < 0.05; ^###^
*p* < 0.001 vs. SB Group. ^¶¶¶^
*p* < 0.001 vs. Cis group.

**Table 5 molecules-25-04688-t005:** Statistical analysis for Bax, Bcl2 and caspase-3 expression in GMSCs.

	Control	G2/SB	Cis	SB and Cis	*p* Value
Bcl-2	25.20 ± 0.51	25.20 ± 0.50	21.52 ± 1.44 ***^###^	17.60 ± 0.27 ***^###¶¶¶^	<0.001
Bax	24.70 ± 1.42	31.48 ± 4.54 **	44.86 ± 0.85 ***^###^	60.42 ± 3.15 ***^###¶¶¶^	<0.001
Caspase 3	21.48 ± 1.32	35.46 ± 2.74 ***	42.96 ± 1.85 ***^###^	55.54 ± 0.76 ***^###¶¶¶^	<0.001

Data expressed as mean ± SD. Test used: one-way ANOVA followed by post-hoc Tukey. ** *p* < 0.01; *** *p* < 0.001 vs. G1/control group. ^###^
*p* < 0.001 vs. G2/SB Group. ^¶¶¶^
*p* < 0.001 vs. G3/Cis group.

**Table 6 molecules-25-04688-t006:** Statistical analysis for Bax, Bcl2 and caspase-3 expression in SCC.

	Control	SB	Cis	SB and Cis	*p* Value
Bcl-2	54.60 ± 2.53	33.54 ± 3.39 ***	29.52 ± 0.28 ***^#^	12.92 ± 1.34 ***^###¶¶¶^	<0.001
Bax	15.60 ± 0.64	43.26 ± 1.48 ***	54.54 ± 1.90 ***^###^	93.48 ± 1.80 ***^###¶¶¶^	<0.001
Caspase 3	14.60 ± 0.067	41.08 ± 0.88 ***	53.40 ± 1.75 ***^###^	83.50 ± 2.48 ***^###¶¶¶^	<0.001

Data expressed as mean ± SD. Test used: one-way ANOVA followed by post-hoc Tukey. *** *p* < 0.001 vs. G1/control group. ^#^
*p* < 0.05; ^###^
*p* < 0.001 vs. G2/SB Group. ^¶¶¶^
*p* < 0.001 vs. G3/Cis group.

**Table 7 molecules-25-04688-t007:** Primer sequences for real time-PCR.

Gene	Forward Primer	Reverse Primer	Product Length (bp)
**BAX**	AGCTGCAGAGGATGATTGCC	CCCCAGTTGAAGTTGCCGTC	100
**BCL2**	TGTGTGTGGAGAGCGTCAAC	CTACCCAGCCTCCGTTATCC	120
**P53**	CATAGTGTGGTGGTGCCCTATGAG	CAAAGCTGTTCCGTCCCAGTAGA	172
**GAPDH**	CTCTGCTCCTCCTGTTCGAC	GCGCCCAATACGACCAAATC	121

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
