# Peer review of "Enhancement of Cisplatin Cytotoxicity by Cu(II)–Mn(II) Schiff Base Tetradentate Complex in Human Oral Squamous Cell Carcinoma"

_molecules, 2020, doi:10.3390/molecules25204688_

Round 1

Reviewer 1 Report

The manuscript by Al-Serwi et al. reports that the heterodinuclear Mn-Cu Schiff base complex, the structure of which is not provided, potentiates cisplatin cytotoxicity. Although the finding is interesting, the rationale to select this particular complex is unclear. In addition, the ligand itself was not evaluated, and it cannot be ascertained whether the effect comes from the ligand or from the entire complex. Because Figure 3 is poorly embedded in the text, only 3a and 3b are visible. Therefore, it is not possible to assess whether the results of MTT assays support conclusions made. It should also be noted that only a narrow concentration range was included, and the error bars in Figure 3 are missing. Based on the MTT data, IC50 values should be calculated and statistical comparison should be performed. The flow cytomretry data should also be statisticallly evaluated. In order to understand the positive effect of the dinuclear complex on cisplatin cytotoxicity, combination index should be determined.

Overall, the fact that only a single compound was studied in one cancer cell line significantly undermines the novelty and the overall merit of the manuscript.

Author Response

Thank you for your comments. Multiple studies have demonstrated mild to good cytotoxicity of Schiff bases against various malignant tumours. Metal inclusion in the complex has a significant impact on the anticancer effectiveness of the Schiff bases. Specifically, Schiff base heterodinuclear Cu (II)–Mn (II) complex is a novel Schiff base metal derivative with reported therapeutic efficacy against liver and colon cancer.

Qin DD, Yang ZY, Zhang FH, Du B, Wang P, Li TR. Evaluation of the antioxidant, DNA interaction and tumour cell cytotoxicity activities of Copper (II) complexes with Paeonol Schiff-base. Inorganic Chemistry Communications. 2010;13(6):727-9.

A clear figure was used to represent MTT results (Figre 3). A log dose response curve was established to calculate IC50 for CIS & Schiff base complex in SCC and GMSCs groups by using graph pad prism version 8.0.  P value less than 0.05 was considered statistically significant. Results are expressed as mean±SD.

Statistical analysis for flow cytometry data was also added Tables 2-7.

Reviewer 2 Report

My review is as follows.   This study deals with the chemosensitizing effect of cisplatin on some SCC cells by a Schiff's base. In the opinion of this reviewer this study would be more meaningful if some initial  cytotoxicity studies using a variety of drugs and a number of different Schiff's bases had been undertaken. The one can undertake detailed studies of the modes of action of one or more promising bioactive compounds.As the situation stands at the moment, one has found a compound with only weak chemosensitizing properties. A second criticism of the manuscript is that it is too verbose such as the introduction which could be reduced by at least 50% and the Discussion section in which much of the first three paragraphs are a repeat of the information in the Introduction section. In order to provide clarity to the cytotoxicity data, a table should be provided with the IC50 values in uM along with standard deviation figures. Data in ug/mL should be converted to uM.  Why was the Schiff's base chosen and not analogs of this compound or other Schiff's bases? A truncated version of this study may be acceptable in this journal.

Author Response

Thank you for your comment. Based on our results, a combination of cisplatin with Schiff’s based produced less anti-proliferative effects and pro-apoptotic effects on GMSCs compared to SCC. Additional studies involving generating the oral SCC xenograft mouse model must be conducted to support our results and to evaluate if the cytotoxic and chemosensitizing effect of Schiff base is outweighing its toxicity on normal cells.

MAJOR REVISIONS

-In the Introduction section, the authors should add some statements and references regarding the role of initiator and effector caspases in the induction of the apoptotic processes, especially for the treatments with cisplatin and Schiff base compounds.

Thank you for your suggestion, we added the role of initiator and effector caspases in the induction of apoptotic process to discussion (Lines 600-604).

-Figure 3 lacks the standard deviations in the graphs of sections a, b, c, d. Please provide the standard deviations for each graphic (n=3 from Materials and Methods) and a Figure of higher quality with all the four subsections clearly visible. Moreover, add a column corresponding to control cells with cell viability conventionally set to 100% and S.D. set to 0 in each of the four graphs.

 Moreover, I do not understand how the authors calculated the IC50 values for cisplatin and the Schiff base compound.  Line 177 Figure 3 shows that the IC50 for cisplatin is higher than 2.5 μg/ml value for both GMSCs and SCCs cell lines. Furthermore, the IC50 of the Schiff base compound seems to be around 600 μM for SCCs cells (Figure 3d) and around 450 μM for GMSCs cells (Figure 3C). This is a critical point of the work, because the Schiff base compound shows a higher cytotoxic effect against normal cells when compared with cancer cells. I think that the authors should use a 150 μM concentration of the Schiff base compound for their experiments, because this concentration possesses cytotoxic effects against SCCs cancer cells with low cytotoxicity against normal GMSCs cells.

In addition, the 150 μM concentration of the Schiff base compound should be used in combination with a cisplatin concentration of about 1.5 μg/ml (the authors indicated the use of a concentration of 1.4 μg/ml).  

A clear figure was used to represent MTT results (Figure 3). A log dose response curve was established to calculate IC50 for CIS & Schiff base complex in SCC and GMSCs groups by using graph pad prism version 8.0.  P value less than 0.05 was considered statistically significant. Results are expressed as mean±SD. The resultant values were 1 µg/ ml for Cisplatin & 250 µM for the Schiff base complex (Lines 108-252).

- In Figure 5, the flow cytometry results show that the Schiff base compound induces apoptosis and necrosis also in GMSCs cells, but the Schiff base and the cisplatin should not kill the normal cells. After the analysis of the graphs of Figure 5, it seems that the authors exchanged the graphs 5F and 5H. In fact, it is reasonable that the pro-apoptotic and pro-necrotic effects of the combination “Schiff  base+Cisplatin” (22.3%+57%=79.3%) are higher than the effects induced by the Schiff compound alone (10.0%+7.3%=17.3%).  If this is not the case, the combination Schiff base+Cisplatin would be nearly useless against these cancer SCCs cells.

We apologize for confusion. Experiments were repeated and alternative figures and tables were added for apoptosis and necrosis assay Figure 4 ,5 &Table 2-5. Although both Schiff base complex and Cisplatin produced apoptosis and necrosis in normal cells which is well established for cisplatin (Florea and Büsselberg , 2011),Cisplatin and Schiff base complex co-treatment may add to sensitizing oral SCC cells to sole treatment. In addition, combination of cisplatin with Schiff’s based produced less anti-proliferative effects and pro-apoptotic effects on GMSCs compared to SCC.

Florea AM, Büsselberg D. Cisplatin as an anti-tumor drug: cellular mechanisms of activity, drug resistance and induced side effects. Cancers (Basel). 2011;3(1):1351-1371. Published 2011 Mar 15. doi:10.3390/cancers3011351

- The authors should also comment the Figure 6 of their Manuscript. Interestingly, Figure 6G shows that the cisplatin treatment induced a G2/M block in the SCCs cancer cells (from 16.3% of control SCCs to 31.1% of SCCs+cisplatin). Moreover, the results of Figure 6D show that the combination of cisplatin+Schiff base increased the sub-G1 population of normal GMSCs cells (from 6.2% of control cells to 11.2%) and decreased the percentage of GMSCs cells in S phase (from 13.7% to 7.6%). How do the authors explain these experimental results?

We apologize for confusion. Experiments were repeated and alternative figures and tables were added for apoptosis and necrosis assay Figure 4 ,5 &Table 2-5.

- Regarding Figure 7, the authors showed that the Schiff base, cisplatin and Schiff base+cisplatin treatments increased the expression levels of the pro-apoptotic genes P53 and BAX and decreased the expression levels of the pro-survival gene BCL2 in SCCs cancer cells. The critical point of the results of Figure 7 is that the same treatments should not increase the expression levels of pro-apoptotic BAX and P53 and should not decrease the expression levels of anti-apoptotic BCL2 in GMSCs cells. In light of these results, the Schiff base and the Schiff base+cisplatin treatments do not possess a specific anticancer effect, because they seem to activate the apoptotic process in both the cancer SCCs cells and normal GMSCs cells. Furthermore, the authors should evaluate if the cisplatin, Schiff base and cisplatin+Schiff base treatments increase the activity levels of initiator caspase 9 and effector caspase 3 in cancer SCCs cells, in order to demonstrate the activation of the intrinsic pathway of apoptosis in these tumor cells.

Thank you for your comment. We added flow cytometry results for BAX, Bcl2 and caspase-3. Both cisplatin and Schiff base complex reduced Bcl2 expression while increased caspase-3 and BAX expression in GMSCs and SCC. Combination of cisplatin and Schiff base complex was more effective than sole treatment. Moreover, effect of treatment on apoptosis markers was more prominent on SCC (Figure 7, 8 and Table 6,7).

- In the Discussion section, the authors wrote: “Herein, gene expression analysis results showed that the used Schiff base complex could efficiently promote cell apoptosis via up-regulating the expression of P53 and Bax proteins and down-regulating Bcl-2 protein”. In order to support this conclusion, the authors must perform western blot experiments to analyze the protein levels of Bax, Bcl-2 and P53 in their cell lines after the treatments with cisplatin, Schiff base and the Schiff base + cisplatin combination. 

Thank you for your suggestion. There are many commercially available kits for Western blotting that can probe different members of proteins. However, intracellular flow cytometry allows for rapid, multiparametric analysis of cells at multiple time points. Also it allows the isolation of specific cell populations based on their cellular membrane protein expression profile and can be used to detect small populations of cells that would otherwise be missed using other techniques because of that we added flow cytometry results for BAX, Bcl2 and caspase-3 (Figure 7, 8 and Table 6, 7).

Reviewer 3 Report

[Molecules] Manuscript ID: molecules-895768

In their Manuscript, Rasha H. Al-Serwi et al. described the cytotoxic and pro-apoptotic effects of a combination of a Schiff base tetradentate complex and cisplatin against the human oral squamous cell carcinoma (SCC) cell line H357. Even if the results are interesting, there is a remarkable problem in their Manuscript: in fact, the authors described also the anti-proliferative effects and  pro-apoptotic effects of the Schiff base and cisplatin against the normal cell line gingival margin-derived stem cells (GMSCs). In order for the combination of these compounds to be used specifically against cancer cells, the authors must demonstrate that their drug combination reduces the proliferation rate of only tumor cells and not also of normal cells. I suggest to replace the GMSCs cells with another oral cancer cell line, like for example human oral squamous carcinoma cell line OEC-M1 and to repeat the  experiments with this new tumor cell line.

MAJOR REVISIONS

-In the Introduction section, the authors should add some statements and references regarding the role of initiator and effector caspases in the induction of the apoptotic processes, especially for the treatments with cisplatin and Schiff base compounds.

-Figure 3 lacks the standard deviations in the graphs of sections a, b, c, d. Please provide the standard deviations for each graphic (n=3 from Materials and Methods) and a Figure of higher quality with all the four subsections clearly visible. Moreover, add a column corresponding to control cells with cell viability conventionally set to 100% and S.D. set to 0 in each of the four graphs. Moreover, I do not understand how the authors calculated the IC50 values for cisplatin and the Schiff  base compound. Figure 3 shows that the IC50 for cisplatin is higher than 2.5 μg/ml value for both GMSCs and SCCs cell lines. Furthermore, the IC50 of the Schiff base compound seems to be around 600 μM for SCCs cells (Figure 3d) and around 450 μM for GMSCs cells (Figure 3C). This is a critical point of the work, because the Schiff base compound shows a higher cytotoxic effect against normal cells when compared with cancer cells. I think that the authors should use a 150 μM concentration of the Schiff base compound for their experiments, because this concentration possesses cytotoxic effects against SCCs cancer cells with low cytotoxicity against normal GMSCs cells. In addition, the 150 μM concentration of the Schiff base compound should be used in combination with a cisplatin concentration of about 1.5 μg/ml (the authors indicated the use of a concentration of 1.4 μg/ml).  

-In Figure 5, the flow cytometry results show that the Schiff base compound induces apoptosis and necrosis also in GMSCs cells, but the Schiff base and the cisplatin should not kill the normal cells. After the analysis of the graphs of Figure 5, it seems that the authors exchanged the graphs 5F and 5H. In fact, it is reasonable that the pro-apoptotic and pro-necrotic effects of the combination “Schiff  base+Cisplatin” (22.3%+57%=79.3%) are higher than the effects induced by the Schiff compound alone (10.0%+7.3%=17.3%).  If this is not the case, the combination Schiff base+Cisplatin would be nearly useless against these cancer SCCs cells.

- The authors should also comment the Figure 6 of their Manuscript. Interestingly, Figure 6G shows that the cisplatin treatment induced a G2/M block in the SCCs cancer cells (from 16.3% of control SCCs to 31.1% of SCCs+cisplatin). Moreover, the results of Figure 6D show that the combination of cisplatin+Schiff base increased the sub-G1 population of normal GMSCs cells (from 6.2% of control cells to 11.2%) and decreased the percentage of GMSCs cells in S phase (from 13.7% to 7.6%). How do the authors explain these experimental results?

- Regarding Figure 7, the authors showed that the Schiff base, cisplatin and Schiff base+cisplatin treatments increased the expression levels of the pro-apoptotic genes P53 and BAX and decreased the expression levels of the pro-survival gene BCL2 in SCCs cancer cells. The critical point of the results of Figure 7 is that the same treatments should not increase the expression levels of pro-apoptotic BAX and P53 and should not decrease the expression levels of anti-apoptotic BCL2 in GMSCs cells. In light of these results, the Schiff base and the Schiff base+cisplatin treatments do not possess a specific anticancer effect, because they seem to activate the apoptotic process in both the cancer SCCs cells and normal GMSCs cells. Furthermore, the authors should evaluate if the cisplatin, Schiff base and cisplatin+Schiff base treatments increase the activity levels of initiator caspase 9 and effector caspase 3 in cancer SCCs cells, in order to demonstrate the activation of the intrinsic pathway of apoptosis in these tumor cells.

- In the Discussion section, the authors wrote: “Herein, gene expression analysis results showed that the used Schiff base complex could efficiently promote cell apoptosis via up-regulating the expression of P53 and Bax proteins and down-regulating Bcl-2 protein”. In order to support this conclusion, the authors must perform western blot experiments to analyze the protein levels of Bax, Bcl-2 and P53 in their cell lines after the treatments with cisplatin, Schiff base and the Schiff base+cisplatin combination. 

Author Response

Thank you for your comments. Multiple studies have demonstrated cytotoxicity of Schiff bases against various malignant tumors. Interestingly, metal inclusion in the complex has a significant impact on the anticancer effectiveness of the Schiff bases. Specifically, Schiff base heterodinuclear Cu (II)–Mn (II) complex is a novel Schiff base metal derivative with reported therapeutic efficacy against liver and colon cancer. Based on this, we have chosen Schiff base heterodinuclear Cu (II)–Mn (II) complex (this was indicated in the introduction (Line 71-75). Dose for Cispaltin and Schiff base complex were selected based on previous studies (Negoro et al 2007 and Neelima et al, 2016), respectively. Introduction and discussion were rephrased to be more focused and repetition was omitted.

Negoro K, Yamano Y, Fushimi K, et al. Establishment and characterization of a cisplatin-resistant cell line, KB-R, derived from oral carcinoma cell line, KB. Int J Oncol. 2007;30(6):1325-1332. doi:10.3892/ijo.30.6.1325.

Neelima, Poonia K, Siddiqui S, Arshad M, Kumar D. In vitro anticancer activities of Schiff base and its lanthanum complex. Spectrochim Acta A Mol Biomol Spectrosc. 2016;155:146-154. doi:10.1016/j.saa.2015.10.015.

Round 2

Reviewer 1 Report

The authors have significantly improved the manuscript. Although this reviewer still that the added value of the work is average, the experiments have been thoroughly performed and analyzed. The manuscript can be accepted after the structure of the complex studied has been included.

Author Response

Thank you for your comments. The structure of Schiff base was added as Supplementary material (1)

Reviewer 3 Report

The authors answered to all my requests and remarkably improved their Manuscript. There are only these two minor revisions left:

1) In the line 252 the authors wrote that "the IC50 values obtained were 1 ug/ml for cisplatin and 250 uM for the Schiff base", but this statement is not clear. Which are the IC50 values for cisplatin in SCC cells and in GMCs cells? Which are the IC50 values for Schiff base complex in SCC and GMCs cells? Please provide all the four IC50 values.

2) In "Figure 6", the Schiff base treatment reduces the mRNA levels of BCL2 of only 10%; please evaluate again the statistical significance of those experiments, because I do not think that the p value can be <0.001. Furthermore, I think that the p value for "SCCs+SB" treatment for BCL2 mRNA levels could be more significant than only p<0.05 (*). Please provide the graph of Figure 6C with the values from 0 to 4 fold changes in the Y axis, so these results can be compared with the results of Figure 6D.

Author Response

The authors answered to all my requests and remarkably improved their Manuscript. There are only these two minor revisions left:

  • In the line 252 the authors wrote that "the IC50 values obtained were 1 ug/ml for cisplatin and 250 uM for the Schiff base", but this statement is not clear. Which are the IC50 values for cisplatin in SCC cells and in GMCs cells? Which are the IC50 values for Schiff base complex in SCC and GMCs cells? Please provide all the four IC50 values.

Thank you for your comment, below are the IC50 for CIS and Schiff base in SCC and GMCs. Based on these values we chose the IC50 for CIS and Schiff in GMCs (the control group).

IC50

CIS-SCC

1.564

CIS-GMCs

1.0206

Schiff-SCC

270.04

Schiff-GMCs

250.01

2) In "Figure 6", the Schiff base treatment reduces the mRNA levels of BCL2 of only 10%; please evaluate again the statistical significance of those experiments, because I do not think that the p value can be <0.001. Furthermore, I think that the p value for "SCCs+SB" treatment for BCL2 mRNA levels could be more significant than only p<0.05 (*). Please provide the graph of Figure 6C with the values from 0 to 4 fold changes in the Y axis, so these results can be compared with the results of Figure 6D.

Thank you very much for your careful revision. The statistics were revised and the graph in figure 6c was adjusted.